# Autistic Traits and Attention-Deficit Hyperactivity Disorder Symptoms Predict the Severity of Internet Gaming Disorder in an Italian Adult Population

**DOI:** 10.3390/brainsci11060774

**Published:** 2021-06-11

**Authors:** Carmen Concerto, Alessandro Rodolico, Chiara Avanzato, Laura Fusar-Poli, Maria Salvina Signorelli, Fortunato Battaglia, Eugenio Aguglia

**Affiliations:** 1Psychiatry Unit, Department of Clinical and Experimental Medicine, University of Catania, 95123 Catania, Italy; c.concerto@policlinico.unict.it (C.C.); chi4493b@gmail.com (C.A.); laura.fusarpoli@gmail.com (L.F.-P.); maria.signorelli@unict.it (M.S.S.); eugenio.aguglia@unict.it (E.A.); 2Department of Medical Sciences, Neurology and Psychiatry, Hackensack Meridian School of Medicine, Nutley, NJ 07110, USA; fortunato.battaglia@hmhn.org

**Keywords:** Autism Spectrum Disorder, Internet Gaming Disorder, problematic internet use, adult, Attention-Deficit Hyperactivity Disorder, online survey, regression analysis, video games

## Abstract

Over the last decade, internet gaming has been a fast-growing recreational activity. Gamers risk their leisure activity becoming an addiction. In the present study, we aimed to measure the prevalence of Internet Gaming Disorder (IGD) in an adult population of video game players and to investigate the association between demographic variables, Autism Spectrum Disorder (ASD) traits, Attention-Deficit Hyperactivity Disorder (ADHD) severity, and IGD in adults. Through an online survey, we recruited 4260 individuals aged between 18 and 55 years old, who were members of online communities of video gamers. We collected demographic data and administered three questionnaires: the Internet Gaming Disorder Scale-Short Form (IGD9-SF), the Autism Spectrum Quotient (AQ), and the Adult ADHD Self-Report Scale (ASRS). Of the overall sample, 29.67% scored above the cut-off of 21 points for the IGD9-SF. Multiple linear regression models showed that daily spare time, autistic traits, and ADHD symptoms were positively associated with the severity of IGD in adults, after controlling for demographic variables. Future studies are required in order to explore factors linked to IGD in adults.

## 1. Introduction

Over the last decade, internet gaming has been a fast-growing recreational activity among adolescents and young adults. The easy, and often free, access to playable content and the ability to become fully immersed in the game have led to an emergent health issue related to excessive online gaming [1]. Previous studies have shown the negative consequences of excessive online gaming in various areas of psychological functioning, potentially resulting in addictive behaviors [2]. Furthermore, online gaming’s instant and constant rewards may profoundly impact the gamers’ attitudes and negatively influence their social lives [3].

In 2013, the American Psychiatric Association (APA) described the phenomenon of Internet Gaming Disorder (IGD) in the fifth edition of the Diagnostic and Statistical Manual of Mental Disorders (DSM-5) [4] as a new condition that requires further study. IGD is characterized by “persistent and recurrent” engagement with video games and a pattern of impaired control over gaming, often causing distress in essential life areas [4]. In spite of the large body of literature, the nosological classification, prevalence, natural history, and the formal treatment of IGD are still a matter of debate [5]. For instance, the prevalence of IGD varies widely, with estimates ranging from 0.21% to 57.5%, depending on the characteristics of the included population and the study recruitment methods [6].

Evidence has shown that IGD, as a non-substance behavioral addiction, is associated with increased feelings of loneliness, decreased social contact, academic dysfunction, substance abuse, poor sleep quality, and complicated relationships with family and peers [1,7]. Emotional regulation has been repeatedly studied as a trait predicting IGD by using different measures. For example, adolescents who play video games show higher levels of alexithymia, which is characterized by difficulties in recognizing and describing emotional feelings and externally oriented thinking [8]. A recent study by Stavropoulos et al. (2019) showed that, among adolescents and young adults, IGD was linked to extreme real-life social withdrawal [9].

Interestingly, common neurodevelopmental disorders, such as Autism Spectrum Disorder (ASD) and Attention-Deficit Hyperactivity Disorder (ADHD), have been associated with higher IGD prevalence compared to the general population [10,11].

ASD is a group of heterogeneous conditions characterized by deficits in social communication and social interactions and by the presence of restrictive and repetitive patterns of behavior, interests, or activities [4]. A recent review [10] included five studies comparing the severity of IGD in individuals with clinical ASD and their neurotypical peers. Higher IGD prevalence was consistently found among individuals with ASD across all age groups. Notwithstanding, one study reported that IGD symptoms did not correlate with parent-reported ASD-like symptoms in autistic children [12]. Only two papers have been published on the subthreshold autistic symptoms as putative risk factors for IGD, with inconsistent findings [13,14].

People with ADHD show a persistent pattern of inattention and/or hyperactivity–impulsivity that interferes with functioning or development [4]. A recent systematic review summarized 29 studies evaluating the association between ADHD and gaming disorder, of which only 11 were based on clinical samples [11]. The review found a consistent positive association between ADHD and IGD, either in clinical-based or community-based samples, particularly for the inattention subscale. In contrast, hyperactivity was less commonly associated with IGD.

It has been theorized that individuals with subthreshold autistic or ADHD traits may be at a greater risk of developing psychiatric conditions than the general population [15,16]. Although the literature has consistently found associations between IGD and both clinical ASD and ADHD, research on subthreshold symptoms in the general adult population is underdeveloped, particularly on autistic-like symptoms. Moreover, to the best of our knowledge, no studies have taken into account the combination of ADHD and ASD traits.

Therefore, we hypothesized that ASD and ADHD subthreshold symptoms may be associated with IGD in an adult population of video game players. The present study was primarily conducted to measure the prevalence of IGD in a population of adult video gamers. Second, we aimed to measure if any sociodemographic variables, ADHD traits, or ASD traits were associated with IGD in adults. This work might have relevant clinical implications, helping to better define the IGD clinical picture in adults, and might help clinicians to deal with specific behavioral patterns in these subjects.

## 2. Materials and Methods

### 2.1. Study Design

We conducted a cross-sectional, web-based observational study between November 2020 and January 2021. Participants were invited via a link to the Google Forms platform. The platform prevented the user from moving to the next section if the previous questions were not completely filled out, avoiding missing data. The survey was designed to be completed in less than 10 min. After the development phase, we piloted the questionnaire internally. Researchers shared the online survey on the web pages of the video gamers’ communities and social networks. The snowball technique was implemented for recruitment.

Eligible individuals included gamers who were members of online communities, aged between 18 and 55 years old, who were able to read and sign the informed consent document. In the online questionnaire, we asked participants whether they had received any psychiatric clinical diagnosis; if the answer was positive, participants were prevented from progressing with the questionnaire. Participation was free, not paid, and was open to anyone who had played at least one online video game during the past year.

The survey was composed of five sections. The first section included information about the project, the study aim, informed consent, and the researchers’ contacts. The second section of the survey included sociodemographic questions. The last three sections of the survey included three standardized questionnaires: the Internet Gaming Disorder Scale-Short Form (IGD9-SF), the Autism Spectrum Quotient (AQ), and the Adult ADHD Self-Report Scale (ASRS).

All of the data were collected anonymously and voluntarily. All participants gave their informed consent. The study was conducted in accordance with the Declaration of Helsinki and was approved by the University of Catania Psychiatry Unit review board (n. 3/2020).

### 2.2. Instruments

The IGD9-SF is a psychometric tool initially developed by Pontes [17]. It is composed of nine items scored on a Likert scale, with possible answers ranging from 1 (never) to 5 (very often). The total score can range between 9 and 45. A recent study identified a cut-off value of 21, in order to distinguish disordered and non-disordered gamers [18]. The scale was validated in the Italian language by Monacis et al. [19].

The AQ is a screening tool designed to identify autistic traits in the general population [20]. It is a self-administered questionnaire composed of 50 Likert-scale items ranging from “Definitely agree” to “Definitely disagree”. Half of the questions are reverse-coded. Using the binary scoring method (the presence of autistic traits, either mildly or strongly, is scored as a +1, while the opposite is scored 0), the total score can range between 0 and 50. Scores of 32 or above may be suggestive of ASD. The AQ was originally developed by Baron-Cohen et al. [21] and validated in the Italian language by Ruta et al. [22].

The ASRS is a screening tool for ADHD in the general population [23]. The ASRS 1.1 version, used in this project, was validated by Silverstain et al. [24]. It was built upon the DSM-5 criteria. The subject is asked to provide responses regarding the frequency of eighteen clinical manifestations, ranging from “Never” to “Very Often”. Depending on the frequency reported, the answer is coded as 1 or 0. The scale covers two sets of questions. Six are considered the best to predict adult ADHD. These questions are “Part A”. If four or more positive answers for this part are given, the subject is considered at high risk of adult ADHD. The other twelve questions consist of “Part B”, which provides additional cues that might be useful in a clinical setting. For this project, we used the scaled overall number of symptoms, namely the sum of “Part A” and “Part B”, which can range from 0 to 18. The Italian version was validated by Somma et al. [25].

### 2.3. Statistical Analysis

All variables were tested for normal distribution before statistical procedures were applied. Continuous variables were presented as means and standard deviations in the case of following the normal distribution. In cases where the data were not normally distributed, medians and interquartile ranges (IQR) were used instead. Categorical variables were presented as percentages and counts. We measured Cronbach’s α for each questionnaire that we employed in our survey in order to evaluate their internal consistency. First, we calculated Spearman’s correlations between the three validated questionnaires. Second, we performed a hierarchical linear regression to evaluate for potential predictors of IGD severity. The IGD scale total score was considered as the dependent variable, while demographics data, the AQ total score, and the ASRS total score were used as predicting variables. Results were considered statistically significant at the *p* < 0.05 level, and all tests were two-tailed. Statistical analyses were performed with SPSS Version 25 [26].

## 3. Results

### 3.1. Sample Characteristics

A total of 4260 individuals ranging from 18 to 55 years of age agreed to participate (median age 26; IQR 23 to 31). Half of the responders lived in Northern Italy (45.05%). The other half were located in Central Italy (22.89%) or Southern Italy and Islands (29.27%). Only a minority of participants were from countries other than Italy (2.79%). Other sociodemographic characteristics are provided in Table 1.

All of the instruments that we used showed acceptable internal consistency. Cronbach’s α values for the IGD9-SF, AQ, and ASRS were 0.78, 0.71, and 0.75, respectively. The overall median score for the IGD scale was 17 (IQR 14 to 21). Of the overall sample, the percentage of subjects above the cut-off of 21 points for the IGD checklist was 29.67%. The overall median score for AQ was 18 (IQR 15 to 23). Of the overall sample, the percentage of subjects above the cut-off of 32 points for the AQ was 2.32%. The median number of ADHD symptoms was 5 (IQR 2 to 7). Of the overall sample, the percentage of subjects above the cut-off of four symptoms for the ASRS “Part A” subscale was 19.27%.

### 3.2. Correlation and Regression Analysis

Results of the correlation between variables included in the regression analysis are reported in Table 2.

The hierarchical regression analysis indicated that there was no collinearity among variables (VIF < 3) [26] (Table 3).

Demographic variables (age, gender, education, relationship, and spare time) were entered into Model 1. The total variance explained by Model 1 was 2.6%, F (54,271) =  23.1, *p*  < 0.0001. Age was inversely associated with a higher IGD scale total score, while having more spare time during the day was positively associated with a higher total score on the IGD scale. The AQ total score (Model 2) explained an additional 9.1% of the total variance, F(64,270)  =  94.7, *p*  < 0.0001. Furthermore, a higher AQ total score was associated with a higher IGD scale score. Lastly, the ASRS total score was entered into the regression (Model 3). Model 3 explained 18.4% of the total variance, F(84,268)  =  120.2, *p*  < 0.0001, and indicated that daily spare time, AQ score, and ASRS score were significant predictors of IGD (Table 3).

## 4. Discussion

This study investigated the association between age, gender, relationship status, daily spare time, AQ total score, ASRS total score, and IGD in a population of adult video gamers. Our results indicate that the prevalence of IGD was 29.67% and that both autistic traits and hyperactivity–inattention symptoms were significantly associated with the severity of IGD in our sample. Our sample prevalence occupies the middle of the general population-wide range identified by Darvesh et al. [6].

To date, the majority of the studies about IGD have been conducted on adolescent samples [27], and young age has been strongly associated with the problematic use of video games in several research works [28].

To the best of our knowledge, we describe for the first time an association between subthreshold autistic traits and IGD, in an adult population of video game players. In recent years, a growing number of studies have emphasized that ASD patients have strong interests in video games, highlighting the potential link between the two phenomena [29]. It has been hypothesized that autistic individuals may be more vulnerable to IGD because of their deficits in social communication and difficulties in building and maintaining real-life relationships. In fact, for autistic individuals, online gaming may represent a more accessible way of communicating with other people. This may mean that this population is at risk for the excessive use of video games [30,31]. Additionally, restricted interests, which represent key features of ASD, may manifest as preoccupations with video games or compulsive patterns of game playing [30]. Of note, autistic traits—i.e., social and communication deficits and repetitive behaviors that fall below the threshold for a diagnosis of ASD—are common among the general population [32]. Our results are in line with a similar study conducted on children, showing that autistic symptoms are linked to IGD through the mediation of emotional regulation and school connectedness [14]. Nevertheless, they are in contrast with Arcelus et al., who found no association between autistic-like traits and IGD in a group of transgender adults [13]. The link between specific ASD subclinical traits and IGD in adults should be further explored considering AQ subscales and additional instruments [33,34].

The association of ASRS score with IGD is consistent with numerous studies indicating that IGD is associated with inattention, hyperactivity, and oppositional behaviors, which are also typical ADHD symptoms in adolescents [35]. Our results further confirm this association in an adult population [36,37]. It was suggested that patients suffering from ADHD show a constant desire for immediate stimulation. Video games’ addictive features and their instant rewards might soothe novelty-seeking in adults with ADHD, becoming a new source of addiction [38]. Furthermore, this conceptual framework is further supported by the fact that ADHD medications reduce IGD symptoms [39]. In terms of neurobiological mechanisms, IGD appears to share characteristics with ADHD. Impaired function in the prefrontal cortex and a deficit in dopamine release have been observed in ADHD patients [5,40]. The prefrontal cortex may be involved in the circuit modulating impulsivity, while its impaired function may be related to high impulsivity, contributing directly to IGD [41].

Evidence has shown that both autistic traits and ADHD symptoms are dimensionally represented in the general population [32,42]. Of note, people with subthreshold symptoms may be more likely to develop psychiatric conditions than the general population [15,16]. As these traits are not clinically manifest and may not cause significant impairments, it might be challenging for mental health professionals to detect them during a formal psychiatric evaluation. Our findings linking ASD traits and ADHD symptoms to IGD may help clinicians to identify and assess the presence of subthreshold ASD or ADHD symptoms in people with IGD. On the other hand, the detection of autistic-like symptoms (e.g., social withdrawal, pervasive interests) and ADHD traits (e.g., inattention, impulsivity) during a psychiatric evaluation may guide professionals to investigate, in-depth, the possible presence of addictive behaviors, such as IGD.

To the best of our knowledge, this is the first study examining the association between IGD, autistic traits and ADHD symptoms among an adult population of video game players. Our results should be interpreted in light of the study’s strengths and limitations. The main strengths are represented by the large sample and by the assessment of IGD symptoms through the IGD9-SF, a validated and methodologically rigorous scale that takes into account the DSM-5 criteria for IGD. The study, however, presents several limitations. It is a cross-sectional study, and we can only report an association between variables. Thus, any discussion about causality is speculative. Longitudinal studies are required in order to explore factors linked to IGD in adults. Another limitation is that the participants in the present work had not received a clinical diagnosis of ADHD and ASD. Thereafter, conducting psychiatric interviews that permit to diagnose the conditions that we investigated would reduce the risk of social desirability or recall bias, which are common when self-report questionnaires are used. Of note, the sample included in the present study was composed of adult video gamers; thus, we cannot generalize our results to the general population. Finally, we did not take into account other psychological variables that may be equally associated with addictive behaviors such as IGD [43,44]. Future studies should include other potential predictors (i.e., temperament, personality traits) or outcomes (i.e., anxiety, depression, sleep disorders) of IGD.

## 5. Conclusions

Our study demonstrated that autistic traits and attention-deficit hyperactivity symptoms are positively associated with the severity of IGD in adult video game players, after controlling for demographic variables. These novel findings may help in designing strategies to identify and prevent the overuse of internet games among adults.

## Figures and Tables

**Table 1 brainsci-11-00774-t001:** Demographic data.

Gender	
Male	3584 (84.13%)
Female	676 (15.87%)
Education level	
Primary	6 (0.14%)
Secondary school	481 (11.29%)
High school	2738 (64.27%)
Bachelor’s degree	623 (14.62%)
Master’s degree	396 (9.30%)
Unknown	16 (0.38%)
Relationship status	
Single	3034 (71.22%)
In a relationship	1226 (28.78%)
Daily spare time	
Less than 1 h	156 (3.66%)
1–2 h	1174 (27.56%)
3–6 h	1822 (42.77%)
More than 6 h	1108 (26.01%)

**Table 2 brainsci-11-00774-t002:** Correlations between psychological variables included in the regression.

Variables	1	2	3
1. IGD total score	1		
2. AQ total score	0.320 **	1	
3. ASRS total score	0.357 **	0.365 **	1

Legend: ** *p* < 0.01.

**Table 3 brainsci-11-00774-t003:** Hierarchical multiple regressions of factors associated with Internet Gaming Disorder Scale-Short Form (IGD9-SF) total score.

Predictor Variables	Model 1 ^a^	Model 2 ^b^	Model 3 ^c^
Age	−0.084 ***	−0.036 *	0.006
Gender	0.01	−0.013	−0.024
Education	−0.037	−0.008	−0.016
Daily spare time	0.112 ***	0.108 ***	0.115 ***
Relationship	−0.029	−0.021	−0.017
AQ total score		0.308 ***	0.211 ***
ASRS total score			0.28 ***
R squared	2.6	11.7	18.4
R squared change	2.6	9.1	6.6
F change	23.1	440.9	345.9

Legend: Standardized coefficient beta; * *p* < 0.05; *** *p* < 0.001; ^a^ Model 1 predictors: age, gender, education, relationship, and spare time; ^b^ Model 2 predictors: Model 1 + AQ total score; ^c^ Model 3 predictors: Model 2 + ASRS total score; AQ: Autism Spectrum Quotient; ASRS: Adult ADHD Self-Report Scale.

## Data Availability

The data are not publicly available.

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
