# Peer review of "Autistic Traits and Attention-Deficit Hyperactivity Disorder Symptoms Predict the Severity of Internet Gaming Disorder in an Italian Adult Population"

_brainsci, 2021, doi:10.3390/brainsci11060774_

Round 1

Reviewer 1 Report

The work presented here is of interest to the scientific community and to the journal's readers. 

However, it also has some considerations in this study. 

First, the title is "Autistic traits and Attention-Deficit Hyperactivity Disorder symptoms predict the severity of Internet Gaming Disorder in adults". The participants were recruited by web-based. The participant has not been diagnosed with a neurodevelopmental disability. Therefore, it is need to change title, for example "Autistic traits and Attention-Deficit Hyperactivity Disorder symptoms predict the severity of Internet Gaming Disorder in an Italian adults population".

Second, despite this study only investigated at trends in autism and ADHD, the authors mostly cite the association between diagnosed of ASD or ADHD and game disorders in their discussion, especially in their discussion of ADHD. If the authors will use these cited literature of discussion, need to add in the Limitation section that participants were not diagnosed.

Third, justify the internal consistency of the measurement in the current study. (ex. The Cronbach’s α coefficient)

Forth, the literature #6 is a bit old, more recent meta-analysis has been published from 2020 or 2021. It would be advisable to update the reference.

Fifth, this is my biggest concern. The age range from 18 to 55 years is too wide in the participants. It is understandable that the age is associated the score of IGD9 from Table 3, however,  the sample's gaming-related characteristics and when participants started playing video games are not helpful from this age range. If this part of the text is to be useful, it should be divided into different age groups and add to create new table, or it does not belong to the study’s main objective, it is recommended that the data be eliminated from the study. 

Author Response

Response to Reviewer 1

1.1 First, the title is "Autistic traits and Attention-Deficit Hyperactivity Disorder symptoms predict the severity of Internet Gaming Disorder in adults". The participants were recruited by web-based. The participant has not been diagnosed with a neurodevelopmental disability. Therefore, it is need to change title, for example "Autistic traits and Attention-Deficit Hyperactivity Disorder symptoms predict the severity of Internet Gaming Disorder in an Italian adults population".

R1.1 We appreciate this comment from the reviewer. We edited the title accordingly.

1.2 Second, despite this study only investigated at trends in autism and ADHD, the authors mostly cite the association between diagnosed of ASD or ADHD and game disorders in their discussion, especially in their discussion of ADHD. If the authors will use these cited literature of discussion, need to add in the Limitation section that participants were not diagnosed.

R1.2 Thanks for suggesting this improvement. We agree with the reviewer that there might be a discrepancy between cited literature on clinical studies and our survey. To underline the fact that our sample was not clinically diagnoses, we have added a brief paragraph among the study limitations:

“Another limitation is that participants of the present work did not receive a clinical diagnosis of ADHD and ASD. Thereafter, conducting psychiatric interviews that permit to diagnose the conditions we investigated would reduce the risk of social desirability or recall bias that are common when self-report questionnaires are used.”

1.3. Third, justify the internal consistency of the measurement in the current study. (ex. The Cronbach’s α coefficient)

R1.3 We thank the reviewer for inviting us to report this relevant information that might increase the validity of our paper. We have reported in the methods section the following sentence: 

“We measured Cronbach’s α of each questionnaire employed in our survey to evaluate their internal consistency.”. 

Then, we added the following sentence in the results paragraph.

“All the instrument we used showed an acceptable internal consistency. Cronbach's α for the IGD9-SF, AQ and ASRS were 0.78, 0.71 and 0.75, respectively” 

1.4 Forth, the literature #6 is a bit old, more recent meta-analysis has been published from 2020 or 2021. It would be advisable to update the reference.

R1.4. We appreciate the author’s suggestion to use a more recent reference. We decided to substitute reference #6 with the following one:

- Darvesh, N., Radhakrishnan, A., Lachance, C. C., Nincic, V., Sharpe, J. P., Ghassemi, M., . . . Tricco, A. C. (2020). Exploring the prevalence of gaming disorder and Internet gaming disorder: a rapid scoping review. Systematic Reviews, 9(1), 68. doi:10.1186/s13643-020-01329-2

1.5. Fifth, this is my biggest concern. The age range from 18 to 55 years is too wide in the participants. It is understandable that the age is associated the score of IGD9 from Table 3, however,  the sample's gaming-related characteristics and when participants started playing video games are not helpful from this age range. If this part of the text is to be useful, it should be divided into different age groups and add to create new table, or it does not belong to the study’s main objective, it is recommended that the data be eliminated from the study. 

R1.5 We thank the reviewer for raising this point. After checking again the overall manuscript we realized that adding gaming related characteristics might be misleading and does not follow exactly the aims of the present project, than we preferred to remove the section from the manuscript. 

Reviewer 2 Report

Review: Autistic traits and Attention-Deficit Hyperactivity Disorder symptoms predict the severity of Internet Gaming Disorder in adults (Concerto et al.)  

The authors performed an online cross section survey investigating relationships between self-reported severity of Internet Gaming Disorder (IGD), autistic traits (Autism Spectrum Quotient), and symptoms of ADHD in adults (ADHD Self Report Scale), in a large sample (n=4260) of Italian Internet gamers, aged 18-55 years. They found significant correlations between the three investigated variables, suggesting that autistic traits and study is interesting because most previous IGD studies focused on adolescents and young adults, and because the results confirm the relationship between IGD, autistic and ADHD symptoms in adult gamers.

The study is comprehensively written, the applied methodologies and statistics are appropriate. There are a few suggestions and comments for improving the quality of the paper.

Specific comments:

Introduction:

Page1/line 38: The authors should add after … (IGD) in the fifth Diagnostic and Statistical Manual of Mental Disorders (DSM5), as a new condition …

Page2: The authors should include two subsections within the introduction: “IGD and ADHD” and “IGD and ASD” to explain why these two disorders predict IGD in children and adolescents.

Please explain why you chose the two comorbid disorders (ADHD & ASD) and not others, such as depression, social anxiety etc.

Please specify the study aims – what are the clinical or practical implications?

Please formulate specific hypotheses.

What are the primary and what are the secondary outcome measures?

Page2/line59: For me the statement “… while predictors of IGD in adults remain largely unknown” is not correct – meanwhile there are studies, e.g.

Rho, Mi & Lee, Hyeseon & Lee, Taek-Ho & Cho, Hyun & Jung, Dong & Kim, Dai-Jin & Choi, Inyoung. (2017). Risk Factors for Internet Gaming Disorder: Psychological Factors and Internet Gaming Characteristics. International Journal of Environmental Research and Public Health. 15. 40. 10.3390/ijerph15010040.

Nurmagandi, B. & Hamid, Achir Yani S. PREDISPOSING ADDICTION FACTOR TO GAME ONLINE:A SYSTEMATIC REVIEW. Vol. 14 No 1 2020 DOI:http://dx.doi.org/10.33533/jpm.v14i1.1654

Sanders, James & Williams, Robert & Damgaard, Marie. (2017). Video Game Play and Internet Gaming Disorder among Canadian Adults: A National Survey. Canadian Journal of Addiction. 8. 6-12. 10.1097/CXA.0000000000000006.

Torez, Miguel (2019). An Online Investigation Into Internet Gaming Disorder (IGD), Comorbidity, and Psychosocial Issues: a Comparison of American and Chinese Gamers—and Predictors of Meeting Criteria for a Formal Diagnosis of IGD. Theses Doctoral. https://doi.org/10.7916/d8-js7c-nx64.

Müller, K., Beutel, M., Egloff, B., & Wölfling, K. (2014). Investigating Risk Factors for Internet Gaming Disorder: A Comparison of Patients with Addictive Gaming, Pathological Gamblers and Healthy Controls regarding the Big Five Personality Traits. European Addiction Research, 20(3), 129-136. Retrieved May 13, 2021, from https://www.jstor.org/stable/26790927

Materials and methods:

Exclusion criteria?

Statistical analysis

Power calculations are missing

Results:

How did the authors deal with missing data?

Page3/line 127ff: Table 1:

  • Within the two variables “daily spare time” and “time spent playing video games” the categories “0-1 hours” and “1-2 hours” are overlapping, 1 hour is in both categories

Page4/line 149: why n has changed? Within Table 2 Legend n=4277, in the text above it is n=4260

Discussion:

Page5/line183-184: This statement is not true, associations between autistic traits and IGD in adult populations have already been described, e.g.:  

  • Problematic internet use and psychiatric comorbidities in a population of Japanese adult psychiatric patients. De Vries et al., 2018
  • Pathological game use in adults with and without Autism Spectrum Disorder. Engelhardt et al., 2017
  • Neural correlates of Non-clinical Internet use in the Motivation Network and its modulation by subclinical autistic traits. Fujiwara et al., 2018
  • Problematic Internet Use and Gaming disorder: a systematic review. Murray et al., 2021

Page6/line215ff: Due to snowball sampling representativeness of the collected sample is limited (potential bias). Generalizability is reduced, because findings don´t include clinical samples, mainly males (84%) and high-school education (64%). Using only self-report instruments there are effects of social desirability or recall bias. Results of self-report questionnaires do not include a diagnosis of ADHD or ASD, so results have to be interpreted cautiously.

References:

The references are cited correctly.

Author Response

Response to Reviewer 2

Introduction:

2.1 Page1/line 38: The authors should add after … (IGD) in the fifth Diagnostic and Statistical Manual of Mental Disorders (DSM5), as a new condition …

R2.1 We thank the reviewer for noticing this missing information. We have now modified the sentence as follows:

In 2013 the American Psychiatric Association (APA) described the phenomenon of Internet Gaming Disorder (IGD) in the fifth edition of the Diagnostic and Statistical Manual of Mental Disorders (DSM-5) as a new condition requiring further studies.”

2.2 Page2: The authors should include two subsections within the introduction: “IGD and ADHD” and “IGD and ASD” to explain why these two disorders predict IGD in children and adolescents.

R2.2 Thank you very much for suggesting this improvement. We agree with the important of presenting the current evidence of the topic. Therefore, we have now added two subsections, as required by the reviewer, to briefly explain the literature published so far. 

ASD is a group of heterogeneous conditions characterized by deficits in social communication and social interactions, and by the presence of restrictive and repetitive patterns of behaviors, interests, or activities [12]. A recent review [10] included five studies comparing IGD severity in individuals with clinical ASD and neurotypical peers. Higher IGD prevalence was consistently found in ASD across all age groups. Notwithstanding, one study reported that IGD symptoms did not correlate with parent-reported ASD-like symptoms in autistic children [13]. Only two papers have been published on the sub-threshold autistic symptoms as putative risk factors for IGD, with inconsistent findings [14,15]. People with ADHD show a persistent pattern of inattention and/or hyperactivity–impulsivity that interferes with functioning or development [12]. A recent systematic review summarized 29 studies evaluating the association between ADHD and gaming disorder, of which only 11 were based on clinical samples [11]. The review found a consistent positive association between ADHD and IGD, either in clinical-based or community-based samples, particularly for the inattention subscale, while hyperactivity was  less commonly associated with IGD. 

The mechanisms underlying the association between IGD and the two conditions under study have been reported in the Discussion.

It has been hypothesized that autistic individuals may be more vulnerable to IGD because of their deficits in social communication and difficulties in building and maintaining real-life relationships. In fact, for autistic individuals, online gaming may represent a more accessible way of communicating with other people. This may make them a population at risk for excessive use of video games [32,33]. Additionally, restricted interests, that represent key features of ASD, may manifest as preoccupations with video games or compulsive patterns of game-playing [32].

[…]

The association of ASRS score with IGD is consistent with numerous studies indicating that IGD is associated with inattention, hyperactivity, and oppositional behaviors, which are also typical ADHD symptoms in adolescents [37].

2.3 Please explain why you chose the two comorbid disorders (ADHD & ASD) and not others, such as depression, social anxiety etc.

R2.3 We thank the reviewer for giving us the opportunity to clarify this issue. Our research group has published several papers on both ASD and ADHD. Therefore, the present study was initially designed because of our interest for neurodevelopmental disorders, a group of conditions that are understudied in adult psychiatry research. Moreover, the literature exploring the association between IGD and subthreshold ASD and ADHD symptoms is still underdeveloped, especially as concern autistic symptoms (see also R2.2). Nevertheless, it has been extensively shown that both autistic traits and ADHD symptoms represent a continuum in the general population and that also people who do not manifest clinical symptoms, but present subthreshold traits, may be at risk to develop psychiatric conditions. As these symptoms are not clinically significant and may not cause impairments, they may remain unrecognized during formal psychiatric evaluations. Vice versa, symptoms such depression and social anxiety are routinely investigated by mental health professionals during clinical interviews and more easily identifiable. Our findings are novel as they may help clinicians suspecting the presence of subthreshold ASD or ADHD symptoms during IGD formal assessment. Nevertheless, we agree that anxiety and depression might be relevant conditions to consider, especially because they might have a mediator role. Thus, we recognize this issue and have suggested to include these variables in future research:

“Finally, we did not take into account other psychological variables that may equally be associated with addictive behaviors such as IGD [36,37]. Future studies should include other potential predictors (i.e. temperament, personality traits) or outcomes (i.e. anxiety, depression, sleep disorders) of IGD.”

2.4 Please specify the study aims – what are the clinical or practical implications? Please formulate specific hypotheses. What are the primary and what are the secondary outcome measures?

R2.4 Thank you for giving us the opportunity to clarify this point. We have now specified the study aims after providing a thorough background and explaining the rationale of our study

“It has been theorized that individuals with subthreshold autistic or ADHD traits may be at a greater risk of developing psychiatric conditions than the general population [16,17]. Although literature has consistently found associations between IGD and both clinical ASD and ADHD, research on subthreshold symptoms in the general adult population is underdeveloped, particularly on autistic-like symptoms. Moreover, to the best of our knowledge, no studies have taken into account the combination of ADHD and ASD traits. Therefore, we hypostasized that ASD and ADHD sub-threshold symptoms may be associated to IGD in adults. The present study was primarily conducted to measure the prevalence of IGD in a population of adult video gamers. Second, we aimed to measure if any sociodemographic variables, ADHD or ASD traits were associated with IGD in adults. This work might have relevant clinical implications to better define IGD clinical picture in adults and might help clinicians deal with specific behavioral patterns of these subjects”

Moreover, we have discussed more in-depth the potential clinical implications of our study in the Discussion section:

Evidence has shown that both autistic traits and ADHD symptoms are dimensionally represented in the general population [33,43]. Of note, people with subthreshold symptoms may be more prone to develop psychiatric conditions than the general population [16,17]. As these traits are not clinically manifest and may not cause significant impairments, it might be challenging for mental health professionals to detect them during a formal psychiatric evaluation. Our findings linking ASD traits and ADHD symptoms to IGD may help clinicians suspect and assess the presence of subthreshold ASD or ADHD symptoms in people with IGD. On the other hand, the detection of autistic-like symptoms (e.g. social withdrawal, pervasive interests), and ADHD traits (e.g. inattention, impulsivity) during a psychiatric evaluation may guide professionals to investigate in-depth the possible presence of addictive behaviors, such as IGD.

2.5 Page2/line59: For me the statement “… while predictors of IGD in adults remain largely unknown” is not correct – meanwhile there are studies, e.g.

- Rho, Mi & Lee, Hyeseon & Lee, Taek-Ho & Cho, Hyun & Jung, Dong & Kim, Dai-Jin & Choi, Inyoung. (2017). Risk Factors for Internet Gaming Disorder: Psychological Factors and Internet Gaming Characteristics. International Journal of Environmental Research and Public Health. 15. 40. 10.3390/ijerph15010040.

- Nurmagandi, B. & Hamid, Achir Yani S. PREDISPOSING ADDICTION FACTOR TO GAME ONLINE:A SYSTEMATIC REVIEW. Vol. 14 No 1 2020 DOI:http://dx.doi.org/10.33533/jpm.v14i1.1654

- Sanders, James & Williams, Robert & Damgaard, Marie. (2017). Video Game Play and Internet Gaming Disorder among Canadian Adults: A National Survey. Canadian Journal of Addiction. 8. 6-12. 10.1097/CXA.0000000000000006.

- Torez, Miguel (2019). An Online Investigation Into Internet Gaming Disorder (IGD), Comorbidity, and Psychosocial Issues: a Comparison of American and Chinese Gamers—and Predictors of Meeting Criteria for a Formal Diagnosis of IGD. Theses Doctoral. https://doi.org/10.7916/d8-js7c-nx64.

- Müller, K., Beutel, M., Egloff, B., & Wölfling, K. (2014). Investigating Risk Factors for Internet Gaming Disorder: A Comparison of Patients with Addictive Gaming, Pathological Gamblers and Healthy Controls regarding the Big Five Personality Traits. European Addiction Research, 20(3), 129-136. Retrieved May 13, 2021, from https://www.jstor.org/stable/26790927

R2.5 This point highlighted by the reviewer is very much welcome because we realize that the sentence was actually misleading. We did not want to say that the psychopathology linked to IGD in adults was missing at all but we wanted to stress the fact that autistic traits and ADHD symptoms are not explored enough in adults video gamers. Then, we completely rephrased this section and removed the misleading sentence.

Materials and methods:

2.6 Exclusion criteria?

R2.6 We thank for pointing this important issue. Indeed there were important missing information in the manuscript: we excluded participants with a known psychiatric diagnosis  as we requested at the beginning of the questionnaire if the subjects had received it, preventing to go further with the questions in case. We are including in the method section the missing information.

In the online questionnaire, we asked participants whether they received any psychiatric clinical diagnosis, preventing participants to go further with the other questions in case the answer was positive.

Statistical analysis

2.7 Power calculations are missing

R2.7 We greatly appreciate the reviewer for asking. Actually we did not do an a priori power analysis, however we have reason to think that this survey sample size is sufficiently large to estimate the prevalence of IGD in adult video gamers in Italy (our primary outcome). Indeed according to a large survey on topic you can find published in Statista.com (specialized website in market and consumer data) (https://www.statista.com/statistics/1111624/number-of-video-game-players-in-italy-by-age-group/) the estimated number of video gamers in Italy is ~10000000. By using the raosoft.com website sample size calculator (http://www.raosoft.com/samplesize.html) with our sample size, we have 95% of chance (error type I) of identifying the correct prevalence of IGD in such a population with a margin of error of 1.5%.

Results:

2.8 How did the authors deal with missing data?

R2.8 We appreciate this question. We clarify that Google Form prevents to go to next session if the previous questions are not completely filled. Thus, we did not have any missing data. We have specified this information in the Methods section

The platform prevented to move to next sections if previous questions were not completely filled, avoiding missing data.

2.9 Page3/line 127ff: Table 1: Within the two variables “daily spare time” and “time spent playing video games” the categories “0-1 hours” and “1-2 hours” are overlapping, 1 hour is in both categories

R2.9 Thanks for pointing at this mistake. Actually the first row should mean “less than 1 hour”, we corrected it accordingly in table 1 regarding “Daily spare time”. The error was present also in the text, referring to “time spent playing video games”. Given the suggestions of the other reviewer we modified the text.

2.10 Page4/line 149: why n has changed? Within Table 2 Legend n=4277, in the text above it is n=4260

R2.10 Thanks for pointing this mistake. It is a typo, we corrected it accordingly.

Discussion:

R2.11 Page5/line183-184: This statement is not true, associations between autistic traits and IGD in adult populations have already been described, e.g.:  

  • Problematic internet use and psychiatric comorbidities in a population of Japanese adult psychiatric patients. De Vries et al., 2018
  • Pathological game use in adults with and without Autism Spectrum Disorder. Engelhardt et al., 2017
  • Neural correlates of Non-clinical Internet use in the Motivation Network and its modulation by subclinical autistic traits. Fujiwara et al., 2018
  • Problematic Internet Use and Gaming disorder: a systematic review. Murray et al., 2021

R2.11 We thank the reviewer for underlining this aspect. As reported in the Introduction and in the review conducted by Murray et al. 2021, five papers specifically evaluated IGD in people with a clinical diagnosis of ASD vs typically developed peers. Conversely, only two papers evaluated the association between autistic-like traits and IGD in the general population. Liu et al. focused on a group of children attending elementary school, while Arcelus et al. evaluated IGD in a group of adult transgenders. This last paper did not find any significant association between autistic-like traits reported at AQ-28 (a shorter version of the AQ) and IGD. Thus, our study is the first to highlight this association in an adult population, and particularly in an adult population of video gamers. Nevertheless, to clarify this aspect, we have modified the sentence as follows:

“To the best of our knowledge, we are describing for the first time an association between subthreshold autistic traits and IGD in an adult population of video game players.”

2.12 Page6/line215ff: Due to snowball sampling representativeness of the collected sample is limited (potential bias). Generalizability is reduced, because findings don´t include clinical samples, mainly males (84%) and high-school education (64%). Using only self-report instruments there are effects of social desirability or recall bias. Results of self-report questionnaires do not include a diagnosis of ADHD or ASD, so results have to be interpreted cautiously.

R2.12 We are grateful for pointing out this relevant limitation that should be remarked in the manuscript. Thereafter we tried to improve the limitations paragraph by adding what follows:

“Another limitation is that participants of the present work did not received a clinical diagnosis of ADHD and ASD. Thereafter, conducting psychiatric interviews that permit to diagnose the conditions we investigated would reduce the risk of social desirability or recall bias that are common when self-report questionnaires are used .”

References:

2.13 The references are cited correctly.

R2.13 Thank you.

Round 2

Reviewer 1 Report

Author correctly response to my review point.